# Development of Alkaline Reduced Water Using High-Temperature-Roasted Mineral Salt and Its Antioxidative Effect in RAW 264.7 Murine Macrophage Cell Line

Thuy Thi Trinh [1,2,†], Ailyn Fadriquela [3,†], Kyu-Jae Lee [1], Johny Bajgai [1], Subham Sharma [1,2], Md. Habibur Rahman [1,2], Cheol-Su Kim [1], Sang-Hum Youn [4] and Hyoung-Tag Jeon [4,5,*]

1 Department of Environmental Medical Biology, Wonju College of Medicine Yonsei University, Wonju 26426, Korea; tththuy@hpmu.edu.vn (T.T.T.); medbio9@gmail.com (K.-J.L.); johnybajgai@gmail.com (J.B.); subhamsharma047@gmail.com (S.S.); pharmacisthabib@gmail.com (M.H.R.); cs-kim@yonsei.ac.kr (C.-S.K.)
2 Department of Global Medical Science, Wonju College of Medicine, Yonsei University, Wonju 26426, Korea
3 Department of Laboratory Medicine, Wonju College of Medicine, Yonsei University, Wonju 26426, Korea; ailyn@yonsei.ac.kr
4 BIOCERA Section, Biocera Co., Ltd., Seongnam-Si 13488, Korea; biornd4@biocera.co.kr
5 Hydrogen Fuel Cell Parts & Applied Technology Regional Innovation Center, Woosuk University, Jeonju 55315, Korea
* Correspondence: jeonht@biocera.co.kr; Tel.: +82-31-6280600
† These authors contributed equally to this study and share first authorship.

**Abstract:** Oxidative stress (OS) plays an important role in many diseases, and its excessive increase affects human health. Although the antioxidant effect of sea salt can be strengthened through special processing, it is scarcely studied. This study confirmed the antioxidative effect of high-temperature roasted mineral salt (HtRMS) produced through repeated roasting of sea salt at high temperature in a ceramic vessel. The dissolved HtRMS exhibited properties such as high alkalinity, rich mineral content, and a high concentration of hydrogen ($H_2$). To detect the antioxidative effect of HtRMS, OS was induced in RAW 264.7 murine macrophage cells with hydrogen peroxide ($H_2O_2$) and lipopolysaccharide (LPS), and then treated with HtRMS solution at different concentrations (0.1, 1, and 10%). Cell viability, reactive oxygen species (ROS), nitric oxide (NO), and antioxidant enzymes such as catalase (CAT) and glutathione peroxidase (GPx), $Ca^{2+}$, and mitogen-activated protein kinase (MAPK) pathway-related proteins (p-p38, p-JNK, and p-ERK) were measured. OS was significantly induced by treatment with $H_2O_2$ and LPS ($p < 0.001$). After treatment with HtRMS, cell viability and GPx activities significantly increased and ROS, NO, $Ca^{2+}$, and CAT significantly decreased in a concentration-dependent manner compared to $H_2O_2$ and LPS-only groups, which was not observed in tap water (TW)-treated groups. Similarly, p-p38, p-JNK, and p-ERK levels significantly decreased in a concentration-dependent manner in HtRMS groups compared to both $H_2O_2$ and LPS-only groups; however, those in TW groups did not exhibit significant differences compared to $H_2O_2$ and LPS-only groups. In conclusion, our results suggest that HtRMS may have antioxidant potential by regulating the MAPK signaling pathway.

**Keywords:** high-temperature-roasted mineral salt; antioxidative effect; MAPK; RAW 264.7

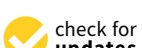



## 1. Introduction

Oxidative stress (OS) represents a pathological imbalance between the antioxidant system and the production of reactive oxygen species (ROS) [1]. ROS, a by-product of metabolism, can participate in some important physiological functions, such as cell signaling, immune functionality, and response to pathogens [2,3]; however, excessive ROS presence leads to cell and tissue injury, leading to OS-related pathological processes [4]. It

is also known that various stimulants such as hydrogen peroxide ($H_2O_2$) and lipopolysaccharide (LPS) can activate ROS-induced signaling pathways [5,6], and that the OS process is modulated by the mitogen-activated protein kinase (MAPK) pathway, which is closely linked to cell proliferation [5].

To combat the detrimental effects of OS-related disorders, various studies and attempts are continuously being conducted, particularly in the nutritional field. This includes several in vivo and in vitro studies on the positive therapeutic effects of mineral salts, such as bamboo salt [7,8]. Cumulative evidence has shown that trace elements, such as calcium ($Ca^{2+}$), potassium ($K^+$), phosphorous ($P^{3-}$), and sulfur ($S^{2-}$), exert antioxidative effects on murine macrophage cell lines [7–9]. Todorka et al. reported that minerals can help control the OS response [10]. Consistent with this, a new concept of mineral salt was developed: the high-temperature-roasted mineral salt (HtRMS) (BIOCERA Co., Ltd., Seongnamsi, Geonggido, Korea), rich in 25 essential minerals and hydrogen ($H_2$), with low oxidation-reduction potential (ORP) values and high alkalinity when dissolved in tap water (TW) [11], is expected to have beneficial effects.

HtRMS contains abundant trace elements such as K (7838 ppm), S (4220 ppm), Ca (1722 ppm), Mg (764 ppm), P (122 ppm), Cl (56.91%), Na (37.94%), Sr (67.8 ppm), I (67 ppm), Si (45.6 ppm), Fe (26.2 ppm), B (18.2 ppm), Li (14.2 ppm), and Be (14.2 ppm). These mineral ions have central roles in cell functions such as cell viability, DNA synthesis, ROS reduction and antioxidative effects [7–9]. $S^{2-}$ typically participates in inflammation and OS and supports the antioxidant system through the formation of glutathione peroxide (GPx) [10]. When dissolved in TW, HtRMS also releases abundant $H_2$, which has known therapeutic potential for OS-related diseases, and has been clinically applied in countries such as Japan, China, the USA, and Korea [12–15]. $H_2$ reportedly exhibits high antioxidant efficiency in medicine [16], likely by inducing an increase in antioxidant enzymes, such as catalase (CAT), superoxide dismutase (SOD), and heme-oxygenase-1 (HO-1) [2]. $H_2$ was experimentally administered in many ways such as drinking, inhaling, injecting, through eye drops, skin smears, and baths [17]. $H_2$ can be produced from the reaction between water and particular compounds such as $Mg^{2+}$ and $Ca^{2+}$, as well as through water electrolysis. HtRMS is completely dissolved in TW and simultaneously produces $H_2$ through an electrochemical reaction with $H_2O$. In addition, the dissolved HtRMS increases the alkalinity of the solution according to a principle similar to that of water electrolysis. The bio-effects of high alkalinity have been extensively studied and reportedly have relevant antioxidative effects [18,19].

Macrophages are key factors in infections and injuries. They are ubiquitous, phenotypically heterogeneous, and have complex functions, including their role in innate and acquired immunity [20], extracellular stimulation, and intracellular responses related to the MAPK signaling pathway [21], which includes phospho-p38 (p-p38), phospho-c-Jun amino-terminal kinases (p-JNK), and phospho-extracellular signal-regulated kinases (p-ERK). These enzymes have diverse biological functions, including nucleosome regulation, gene expression, mRNA stabilization and translation, and cell proliferation and survival [22].

In this study, the antioxidative effect of a fortified alkaline water was tested using HtRMS, which was produced through repeated roasting at a high temperature, in a murine macrophage cell line exposed to different stress stimulants, such as $H_2O_2$ and LPS, and compared with TW effects. We also analyzed the potential mechanism of action of HtRMS by examining its effects on the MAPK signaling pathway.

## 2. Materials and Methods

### 2.1. Experimental Materials

Dulbecco's modified Eagle's medium (DMEM) was purchased from Cytiva Hyclone (South Logan, OH, USA). The penicillin/streptomycin (antibiotics) solution was purchased from GibcoTM (Invitrogen Corporation, Auckland, NY, USA); fetal bovine serum (FBS) was purchased from Hyclone Laboratories, Inc. (South Logan, OH, USA). LPS (serotype

O111:B4) was purchased from Sigma-Aldrich (St. Louis, MO, USA), and 30% $H_2O_2$ was purchased from Daejung (Siheung-si, Gyeonggi-do, Korea). A Cell Counting kit-8 (CCK-8) was purchased from Quanti-MaxTM. The CCK-8 assay kit was purchased from Dojindo Molecular Technologies (Rockville, MD, USA); 2′7′-dicholodihydrofluorescein diacetate (DCFH-DA) reagent was purchased from Sigma Chemical Co. (Sigma, St. Louis, MO, USA); nitric oxide (NO) reagent (Griess reagent kit) was purchased from iNtRON Biotechnology (Sungnam, South Korea); a CAT assay kit, GPx assay kit, and $Ca^{2+}$ colorimetric assay kit was obtained from BioVision Inc. (Milpitas, CA, USA). The TakaRa BCA protein assay kit was obtained from Takara Bio Inc. (Shimadzu, Tokyo, Japan). Antibodies recognizing phospho-p38 (p-p38), phospho-c-Jun N-terminal kinase (p-JNK), phospho-extracellular signal-related kinase (p-ERK) 1 and 2 and β-actin (dilution 1:2000) were rabbit monoclonal IgM, whereas secondary antibodies (dilution 1:5000) were horseradish peroxidase-linked anti-rabbit IgG, all obtained from Cell Signaling Technology (Danvers, MA, USA).

Mineral Contents and Properties of HtRMS

The experimental material, HtRMS (Hydrogen A.A Mineral Salt, BIOCERA Co., Ltd., Seongnamsi, Geonggido, Korea), is water soluble and was produced through a high-temperature roasting process at 750–800 °C for 30 min in a bio-ceramic jar (diameter 15 cm, height 15 cm, thickness 1 cm), this process was repeated 10 times and then the final salt was pulverized in the size of 0.1–0.2 mm. The bio-ceramic jar used in the experiment was made through a particular manufacturing process including a high-temperature firing process at 900–1250 °C for 1 h. The mineral content (Table 1) and properties (Table 2) of HtRMS were analyzed.

**Table 1.** Mineral component content analysis of HtRMS.

| Mineral Components | Contents (ppm) | Mineral Components | Contents (ppm) |
|---|---|---|---|
| Calcium ($Ca^{2+}$) | 1722 | Copper ($Cu^{2+}$) | 2.28 |
| Phosphorous ($P^{3-}$) | 122 | Barium ($Ba^{2+}$) | 6.46 |
| Potassium ($K^+$) | 1738 | Tin ($Sn^{4+}$) | 1.24 |
| Sulfur ($S^{2-}$) | 4220 | Iodine ($I^-$) | 67 |
| Sodium ($Na^+$) | 379,400 | Titanium ($Ti^{3+}$) | 2.6 |
| Chlorine ($Cl^-$) | 569,600 | Boron ($B^{3+}$) | 18.2 |
| Magnesium ($Mg^{2+}$) | 746 | Selenium ($Se^{2-}$) | 3.01 |
| Ion ($Fe^{2+}$) | 26.2 | Lithium ($Li^+$) | 14.2 |
| Fluorine ($F^-$) | 18 | Molybdenum ($Mo^{2+}$) | 1.95 |
| Zinc ($Zn^{2+}$) | 3.57 | Gallium ($Ga^{3+}$) | 10.36 |
| Silicon ($Si^{4+}$) | 45.6 | Vanadium ($V^-$) | 9.73 |
| Rubidium ($Rb^+$) | 6.27 | Beryllium ($Be^{2+}$) | 14.2 |
| Strontium ($Sr^+$) | 67.8 | Bromine ($Br^-$) | 8 |

Component analysis was conducted in the Hydrogen Fuel Cell Parts & Applied Technology Regional Innovation Center, Woosuk University, Wanju, Republic of Korea. HtRMS: high-temperature-roasted mineral salt.

**Table 2.** Properties of HtRMS.

| Materials | pH | ORP (mV) | TDS (ppm) | $H_2$ (ppb) |
|---|---|---|---|---|
| TW | 7.32 | 609 | 105 | 0 |
| HtRMS | 10.5 | −380 | 27,000 | 450 |

TW: tap water; HtRMS: high-temperature-roasted mineral salt; ORP: oxidation-reduction potential; TDS: total dissolved solid.

### 2.2. Experimental Design

RAW 264.7 murine macrophage cell line (American Type Cell Culture Collection, Manassas, VA, USA) was cultured and treated for 7 d according to the experimental protocol (Figure 1). The cells were cultured for 5 d and then treated with 200 µM/mL $H_2O_2$ for 2 h and 10 µg/mL LPS for 24 h to induce OS. The induction of OS was confirmed by

detecting cell viability compared to that in untreated. OS-induced cells were treated with different concentrations (0.1, 1, and 10%) of HtRMS (HtRMS-treatment group) and TW (TW-treatment group) for 24 h. On the 7th day of the experiment, the cells were collected, and OS-related biomarkers such as ROS, NO, CAT, GPx, and $Ca^{2+}$ were measured. MAPK signaling pathway proteins such as p-p38, p-JNK, and p-ERK were also evaluated via Western blotting.

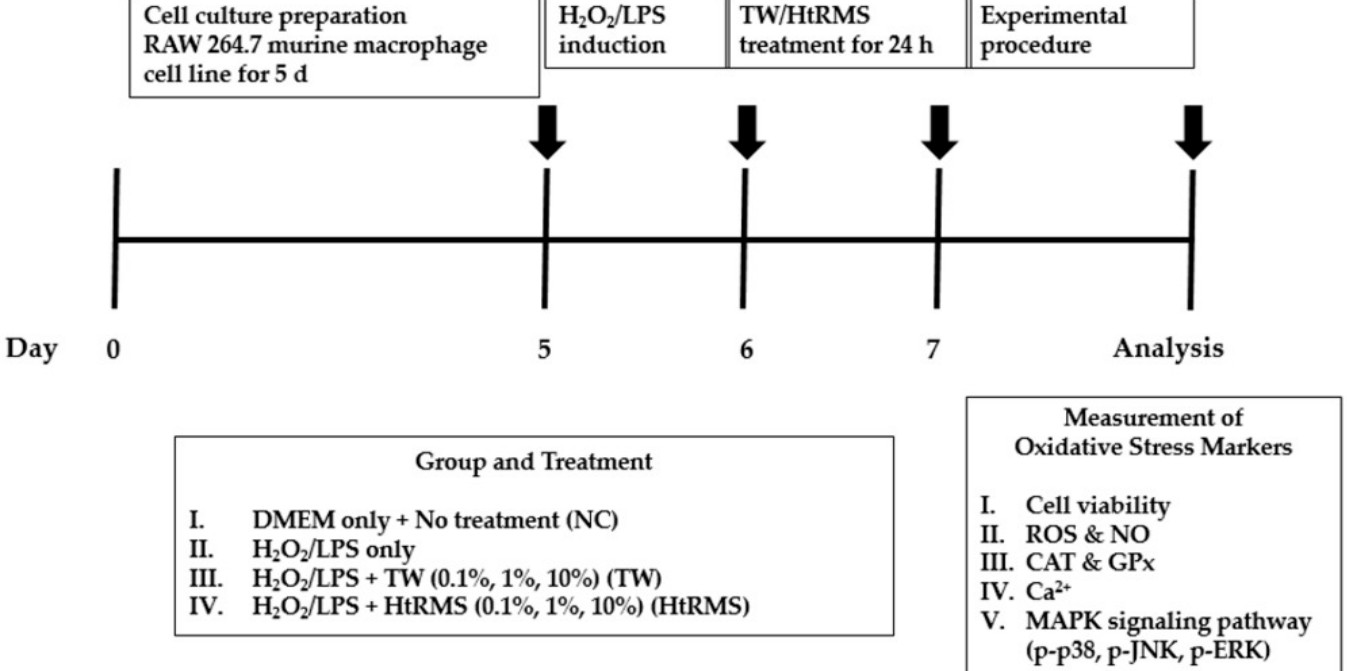

**Figure 1.** Outline of the experimental procedure. ROS: reactive oxygen species; NO: nitric oxide; CAT: catalase; GPx: glutathione peroxidase; MAPK: mitogen-activated protein kinase; $H_2O_2$: hydrogen peroxide; LPS: lipopolysaccharide.

### 2.3. Cell Culture and LPS Stimulation

To identify the ideal concentration of LPS its stimulation time, RAW 264.7, a macrophage cell line, was grown in DMEM with 10% FBS and 1% antibiotic and cultured in 75 mm flasks at 37 °C with 5% $CO_2$ in a humidified atmosphere. RAW 264.7 cells ($5 \times 10^3$ cells/well) were used in these experiments. The cells were treated with different concentrations of LPS (0.4, 2, and 10 µg/mL) for 6, 12, 24, and 48 h after reaching 80% confluence. Cell viability was evaluated using the CCK-8 assay kit, following the manufacturer's protocol. Finally, the results were analyzed by $IC_{50}$ to select the ideal concentration and time of LPS stimulation to induce OS.

### 2.4. Cell Culture and $H_2O_2$ Stimulation

To identify the ideal concentration and time of $H_2O_2$ stimulation, RAW 264.7 cells ($5 \times 10^3$ cells/well) were treated with 50, 100, and 200 µM/mL of $H_2O_2$ for 30, 60, 90, and 120 min. Cell viability was measured using the CCK-8 assay kit, following the manufacturer's protocol. Similarly, the result was analyzed by $IC_{50}$ to determine the ideal concentration and time of $H_2O_2$ stimulation to induce OS.

### 2.5. Cell Proliferation Assay

CCK-8 was used to evaluate cell viability following the manufacturer's protocol. Briefly, RAW 264.7 macrophage cells ($1 \times 10^4$ cells/well) were seeded in a 96-well plate and incubated at 37 °C in 5% $CO_2$ for 24 h. After washing 2 times with PBS $1\times$, the cells were treated with LPS or $H_2O_2$ for the indicated time and concentration. Subsequently, the cells were treated with HtRMS (0.1, 1, and 10%, respectively) for 24 h. Briefly, 10 µL/well CCK-8

was transferred to each well, and the cells were incubated for 2 h at 37 °C. An absorbance microplate reader (Molecular Devices, San Jose, CA, USA) was used to measure the optical density of each well at 380 nm.

### 2.6. ROS Assay

DCFH-DA reagent was used to determine intracellular ROS levels following the manufacturer's instructions. RAW 264.7 cells ($1 \times 10^4$ cells/well) were seeded in a 96-well blank plate before treatment with HtRMS. Afterward, the cells were washed twice with PBS and placed in a mixture of 20 μL lysis buffer and 30 μL PBS 1×. Finally, 100 μL of 10 μM DCFH-DA was added to each well and incubated for 30 min at 37 °C. A DTX multi-mode micro plate reader (Beckman Coulter Inc., Brea, CA, USA) was used to measure the fluorescence at 488 nm excitation/525 nm emission.

### 2.7. NO Assay

Griess reagent (iNtRON) was used to measure NO. The cells were seeded in 96-well plates ($1 \times 10^4$ cells/well) and treated with HtRMS. Subsequently, the cells were washed twice with PBS 1× and replaced with a mixture of 20 μL lysis buffer and 30 μL PBS 1×. Finally, following the manufacturer's protocol, Griess reagent was added to all wells and incubated at RT for 15 min. An absorbance microplate reader (Molecular Devices, CA, USA) was used to measure the absorbance of each well at 540 nm.

### 2.8. Endogenous Antioxidant Enzyme Activities

The endogenous antioxidant enzymes (CAT and GPx) were evaluated using the BioVision kit (Milpitas, CA, USA). The cells were seeded into 6-well plates ($0.2 \times 10^6$ cells/well) and then lysed by assay buffer before centrifugation at 10,000 rpm for 15 min at 4 °C. The cell supernatant was used to measure CAT and GPx activities according to the manufacturer's instructions. An absorbance microplate reader (Molecular Devices, CA, USA) was used to measure the optical density of CAT (570 nm) and GPx (340 nm).

### 2.9. $Ca^{2+}$ Assay

Intracellular $Ca^{2+}$ levels were measured using a colorimetric assay kit (BioVision, Milpitas, CA, USA). The cells were seeded into 100 mm culture dishes ($2 \times 10^4$ cells/well). After collection, the cells were lysed using the $Ca^{2+}$ assay buffer and centrifuged at 10,000 rpm for 10 min at 4 °C. $Ca^{2+}$ levels in the supernatants were measured according to the manufacturer's protocol. Samples were added to the calcium standards; the reaction mix was added into a 96-well microplate, which was then incubated. An absorbance microplate reader (Molecular Devices, San Jose, CA, USA) was used to measure the optical density at 575 nm.

### 2.10. Western Blot Analysis

After normalizing the protein concentration, the cell supernatant was loaded and separated via SDS-polyacrylamide gel electrophoresis. The protein bands in the gel were electrophoretically transferred to polyvinylidene difluoride membranes (Sartorius, Bohemia, NY, USA). The membrane was blocked with blocking buffer (Takara Bio Inc., Shiga, Japan) at room temperature (RT) for 2 h and incubated with the following primary antibodies: p-p38, p-JNK, p-ERK, and β-actin (dilution: 1:2000; Cell Signaling Technology, Danvers, MA, USA) in Tris-buffered saline/Tween 20 (TBS-T 1×) containing 5% bovine serum albumin overnight at 4 °C. The secondary antibody used was anti-rabbit (dilution 1:2000; Cell Signaling Technology) and incubated at RT for 2 h. Antibodies were detected via chemiluminescence (ECL Pierce Biotechnology) UVP Bio spectrum 600 Imaging System (UVP, LLC, Upland, CA, USA). β-actin (dilution 1: 2000, Cell Signaling Technology) was used as a loading control for total protein content. Band intensity was analyzed using ImageJ software (Version 150-win Java, USA).

*2.11. Data Management and Statistical Analysis*

Data standards were taken with the mean value ± standard error of the mean (SEM). All data in each marker were normalized, and fold change was computed according to normal control and were analyzed and compared by one-way analysis of variance (ANOVA) followed by a multiple comparison test with GraphPad Prism 8.0 software package (GraphPad, La Jolla, CA, USA). Differences were considered statistically significant at $p < 0.05$.

## 3. Results

*3.1. Effect of HtRMS on $H_2O_2$- and LPS-Induced Cell Viability of Murine Macrophage RAW 264.7 Cells*

Cell viability was significantly reduced after treatment with $H_2O_2$ ($p < 0.001$) and LPS ($p < 0.001$) compared to the normal control (NC) (Figure 2A,B). Upon HtRMS treatment, cell viability significantly increased in the HtRMS 10% group ($p < 0.001$) compared to the $H_2O_2$-only group, whereas cell viability in the TW group decreased significantly at 1% ($p < 0.01$) and 10% ($p < 0.001$) as compared to the $H_2O_2$-only group (Figure 2A). The induction of LPS also exhibited a similar trend, wherein the cell viability of the HtRMS 10% group significantly increased ($p < 0.01$) (Figure 2B).

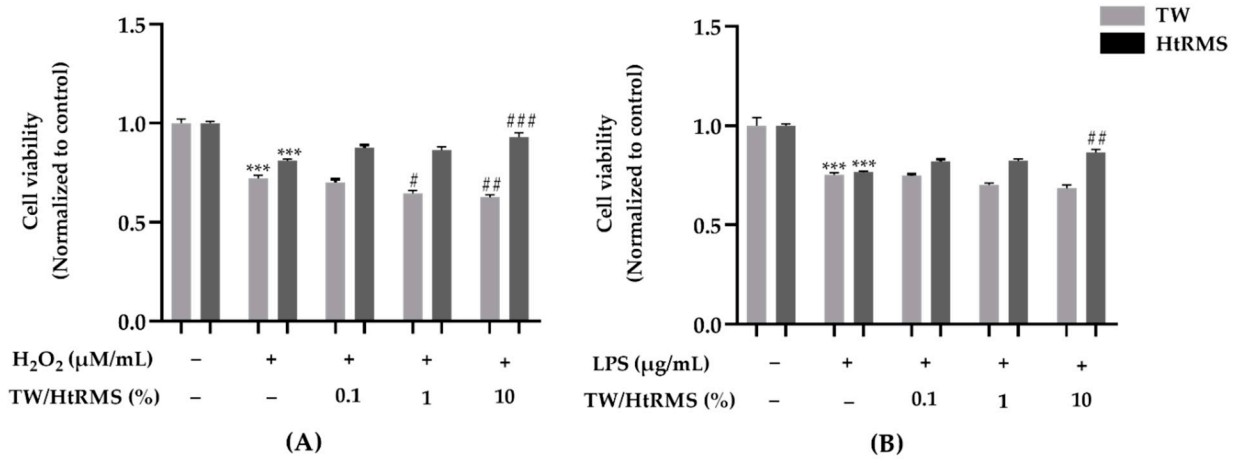

**Figure 2.** Effect of HtRMS on cell viability of $H_2O_2$- and LPS-induced murine macrophage cells. The cell toxicity was induced by $H_2O_2$ (**A**) and LPS (**B**) and then treated with TW and HtRMS, respectively, in different concentrations of 0.1, 1, and 10%. Data values are stated as mean ± standard error of the mean of fold change according to control ($n = 3$). The statistical significance was analyzed via two-way ANOVA. TW: tap water; (−): non-treatment; (+): treatment; ***: $p < 0.001$ vs. normal control; #: $p < 0.05$, ##: $p < 0.01$, and ###: $p < 0.001$ vs. $H_2O_2$ or LPS only. HtRMS: high-temperature-roasted mineral salt; TW: tap water; $H_2O_2$: hydrogen peroxide; LPS: lipopolysaccharide.

*3.2. Effect of HtRMS on OS Production of $H_2O_2$- and LPS-Induced Murine Macrophage RAW 264.7 Cells*

ROS and NO levels significantly increased compared to the NC group after induction by $H_2O_2$ ($p < 0.001$) and LPS ($p < 0.001$) (Figure 3). After HtRMS treatment, ROS decreased in a concentration-dependent manner in HtRMS 0.1% ($p < 0.05$), 1% ($p < 0.001$), and 10% ($p < 0.001$) groups compared to the $H_2O_2$-only group; however, ROS in the TW groups increased significantly at 1% ($p < 0.01$) and 10% ($p < 0.001$) concentrations (Figure 3A). The induction by LPS also exhibited a similar trend to that by $H_2O_2$, and ROS levels significantly decreased in the HtRMS 10% group ($p < 0.01$) compared to LPS-only group (Figure 3B). In addition, NO drastically decreased in a concentration-dependent manner at 0.1, 1, and 10% concentrations ($p < 0.001$) in the HtRMS groups compared to that in the $H_2O_2$-only group (Figure 3C). Similarly, after LPS induction, NO significantly decreased in the HtRMS 1% ($p < 0.05$) and 10% ($p < 0.001$) groups, and there was no significant difference in the TW groups (Figure 3D).

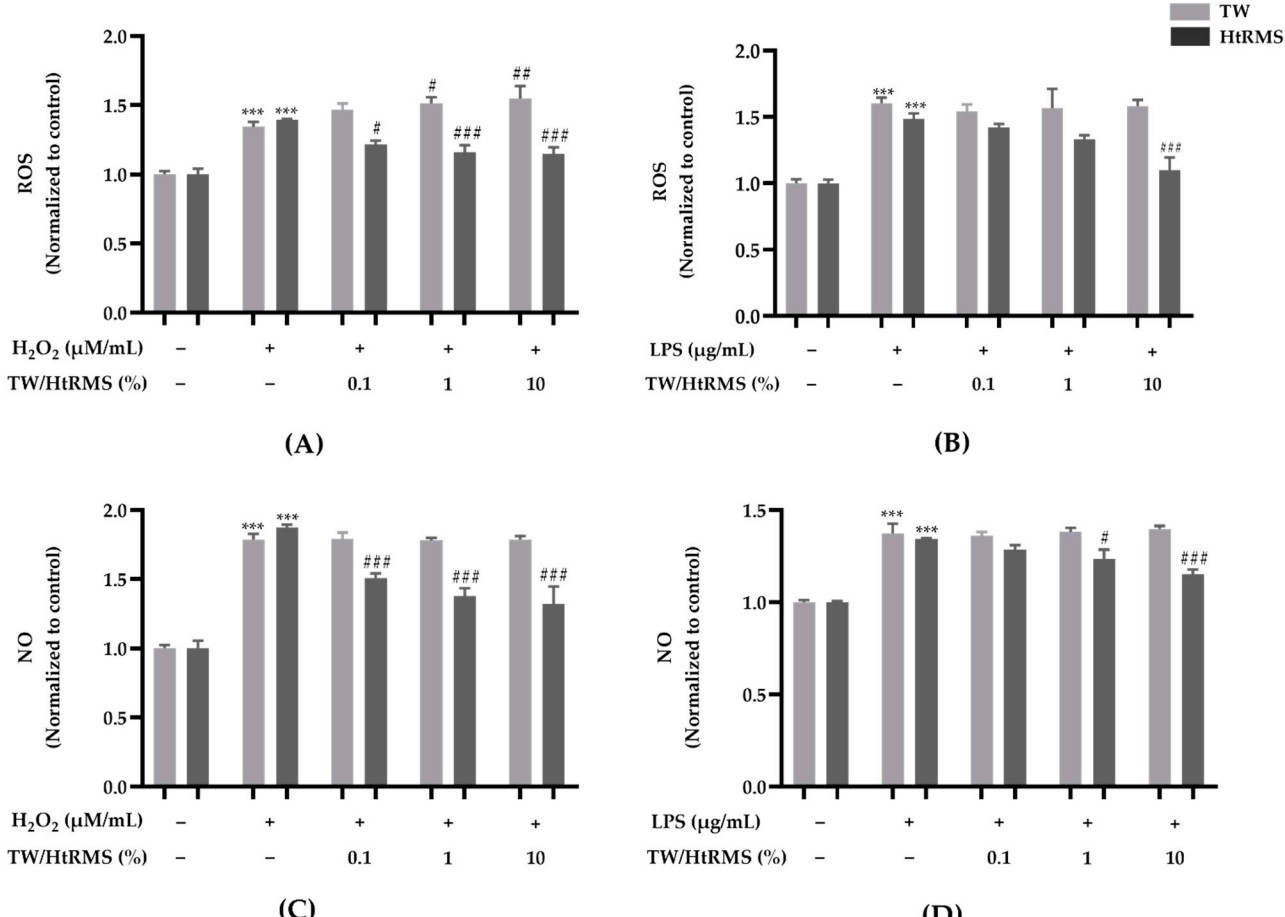

**Figure 3.** Effect of HtRMS on OS production of $H_2O_2$- and LPS-induced murine macrophage cells. Table 0; 1 and 10% concentration after OS induction by $H_2O_2$ (**A**,**C**) and LPS (**B**,**D**), respectively. Data values are stated as mean $\pm$ standard error of the mean of fold change relative to the control group ($n = 3$). The statistical significance was analyzed via two-way ANOVA. ($-$): non-treatment; (+): treatment. ***: $p < 0.001$ vs. normal control; #: $p < 0.05$; ##: $p < 0.01$, and ###: $p < 0.001$ vs. $H_2O_2$ or LPS only. HtRMS: high-temperature-roasted mineral salt; TW: tap water; $H_2O_2$: hydrogen peroxide; LPS: lipopolysaccharide.

### 3.3. Effect of HtRMS on Intracellular Antioxidant Enzyme Levels of $H_2O_2$- and LPS-Induced Murine Macrophage RAW 264.7 Cells

CAT levels significantly increased after induction by $H_2O_2$ and LPS compared to that in the non-treated group ($p < 0.001$) (Figure 4A,B). After HtRMS treatment, CAT significantly decreased in a concentration-dependent manner at 0.1% ($p < 0.05$), 1% ($p < 0.01$), and 10% ($p < 0.001$) compared to the $H_2O_2$-only group (Figure 4A); however, the TW group exhibited no significant difference. LPS-induced cells also exhibited a similar trend to the one consequent to $H_2O_2$ induction. CAT significantly decreased in the HtRMS 0.1% ($p < 0.05$), 1% ($p < 0.001$), and 10% ($p < 0.001$) groups compared to the LPS-only group (Figure 4B). In contrast, GPx drastically increased in the HtRMS 1% ($p < 0.001$) and 10% ($p < 0.001$) groups compared to the $H_2O_2$-only group (Figure 4C) and to the LPS-only group ($p < 0.01$ and $p < 0.001$, respectively) (Figure 4D).

### 3.4. Effect of HtRMS on the Intracellular $Ca^{2+}$ Level of $H_2O_2$- and LPS-Induced Murine Macrophage RAW 264.7 Cells

$Ca^{2+}$ activity significantly increased after the induction of $H_2O_2$ and LPS compared to that in the non-treated group ($p < 0.001$) (Figure 5A,B). Upon HtRMS treatment, $Ca^{2+}$ levels decreased significantly at 0.1% ($p < 0.01$), 1% ($p < 0.001$), and 10% ($p < 0.001$) concentrations compared to that in the $H_2O_2$-only group (Figure 5A), and similarly at 1% ($p < 0.001$) and

10% ($p < 0.001$) concentrations compared to that in LPS-only group (Figure 5B); however, TW treatment groups did not exhibit significant differences.

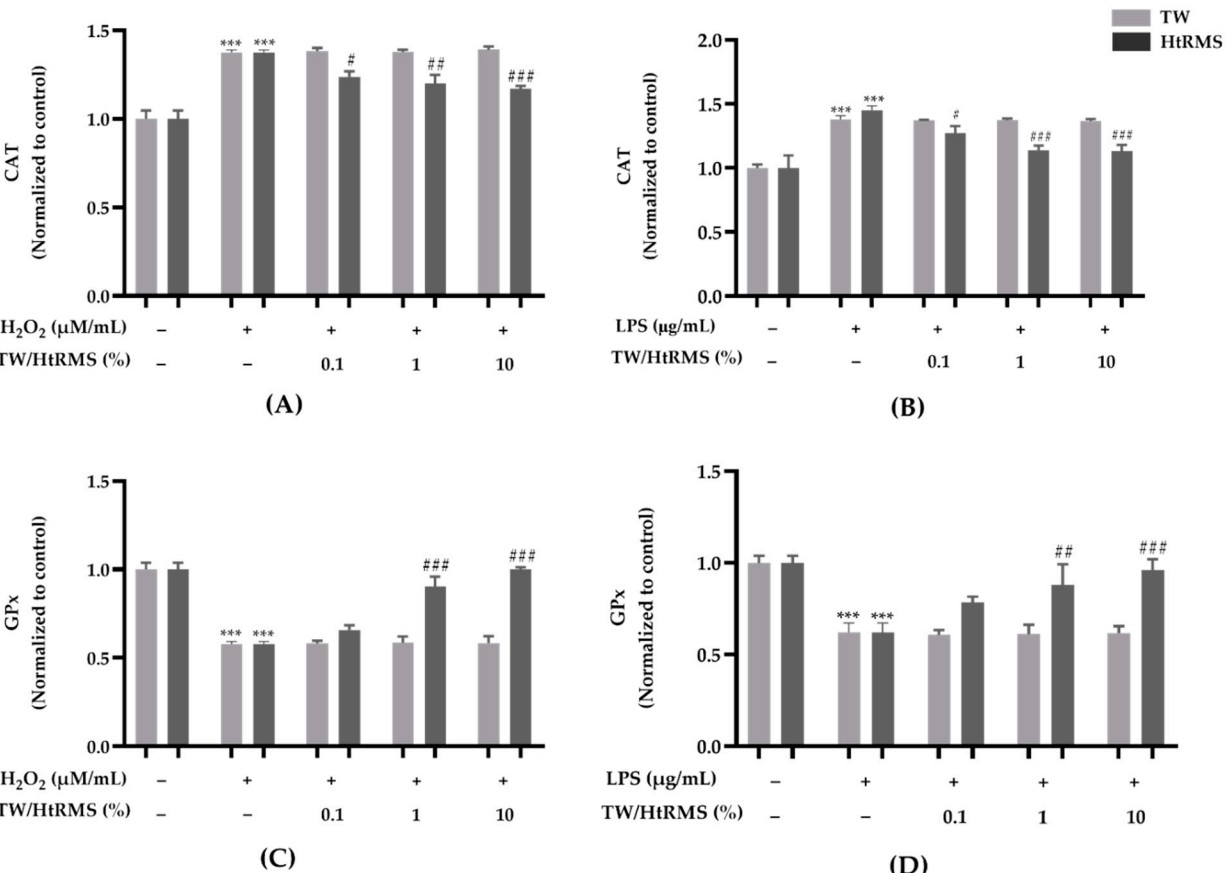

**Figure 4.** Effects of HtRMS on the levels of CAT (**A**,**B**) and GPx (**C**,**D**) of $H_2O_2$- and LPS-induced murine macrophage cells. The cells were treated with HtRMS and TW (0.1, 1, and 10%) after OS induction by $H_2O_2$ (**A**,**C**) and LPS (**B**,**D**), respectively. Data values are stated as mean ± standard error of the mean of fold change according to the control group ($n = 3$). The statistical significance was analyzed via two-way ANOVA. (−): non-treatment; (+): treatment. ***: $p < 0.001$ vs. normal control. #: $p < 0.05$, ##: $p < 0.01$, and ###: $p < 0.001$ vs. $H_2O_2$- or LPS-only. HtRMS: high-temperature-roasted mineral salt; TW: tap water; $H_2O_2$: hydrogen peroxide; LPS: lipopolysaccharide; CAT: catalase, GPx: glutathione peroxidase.

### 3.5. Effect of HtRMS on p-p38, p-JNK, and p-ERK of $H_2O_2$- and LPS-Induced Murine Macrophage RAW 264.7 Cells

p-p38, p-JNK, and p-ERK significantly increased after $H_2O_2$ and LPS induction compared to the NC group ($p < 0.001$) (Figure 6A,B). After HtRMS treatment, p-p38, p-JNK, and p-ERK exhibited a significantly decreasing trend in a concentration-dependent manner compared to the $H_2O_2$-only groups; however, the TW group did not exhibit any significant difference. p-p38 expression increased at 1% ($p < 0.01$) and 10% ($p < 0.001$) treatments, p-JNK at 1% ($p < 0.05$) and 10% ($p < 0.001$) treatments, and p-ERK at 0.1% ($p < 0.01$), 1% ($p < 0.001$), and 10% ($p < 0.001$) treatments compared to that in the $H_2O_2$-only groups (Figure 6A). Similarly, in LPS-induced cells, p-p38, p-JNK, and p-ERK significantly decreased at any HtRMS concentration ($p < 0.001$) compared to LPS-only groups; however, they significantly increased in the TW groups compared to LPS-only groups (Figure 6B).

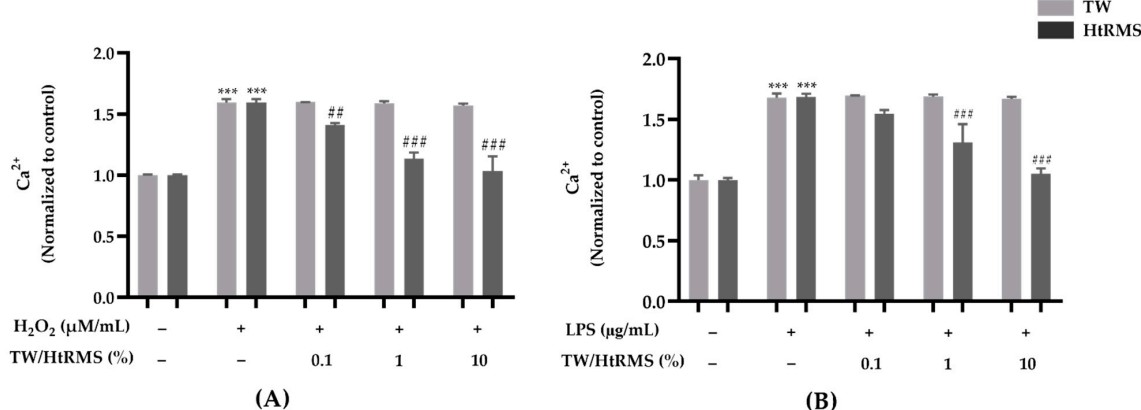

**Figure 5.** Effects of HTRMS on $Ca^{2+}$ activity of $H_2O_2$ and LPS-induced murine macrophage cells. The cells were treated with HtRMS and TW (0.1, 1, and 10%), after the induction of OS by $H_2O_2$ (**A**) and LPS (**B**), respectively. Data values are stated as mean $\pm$ standard error of the mean of fold change compared to the control group ($n = 3$). The statistical significance was analyzed via two-way ANOVA. ($-$): non-treatment, (+): treatment. ***: $p < 0.001$ vs. normal control. ##: $p < 0.01$ and ###: $p < 0.001$ vs. $H_2O_2$ or LPS only. HtRMS: high-temperature-roasted salt; TW: tap water; $H_2O_2$: hydrogen peroxide; LPS: lipopolysaccharide.

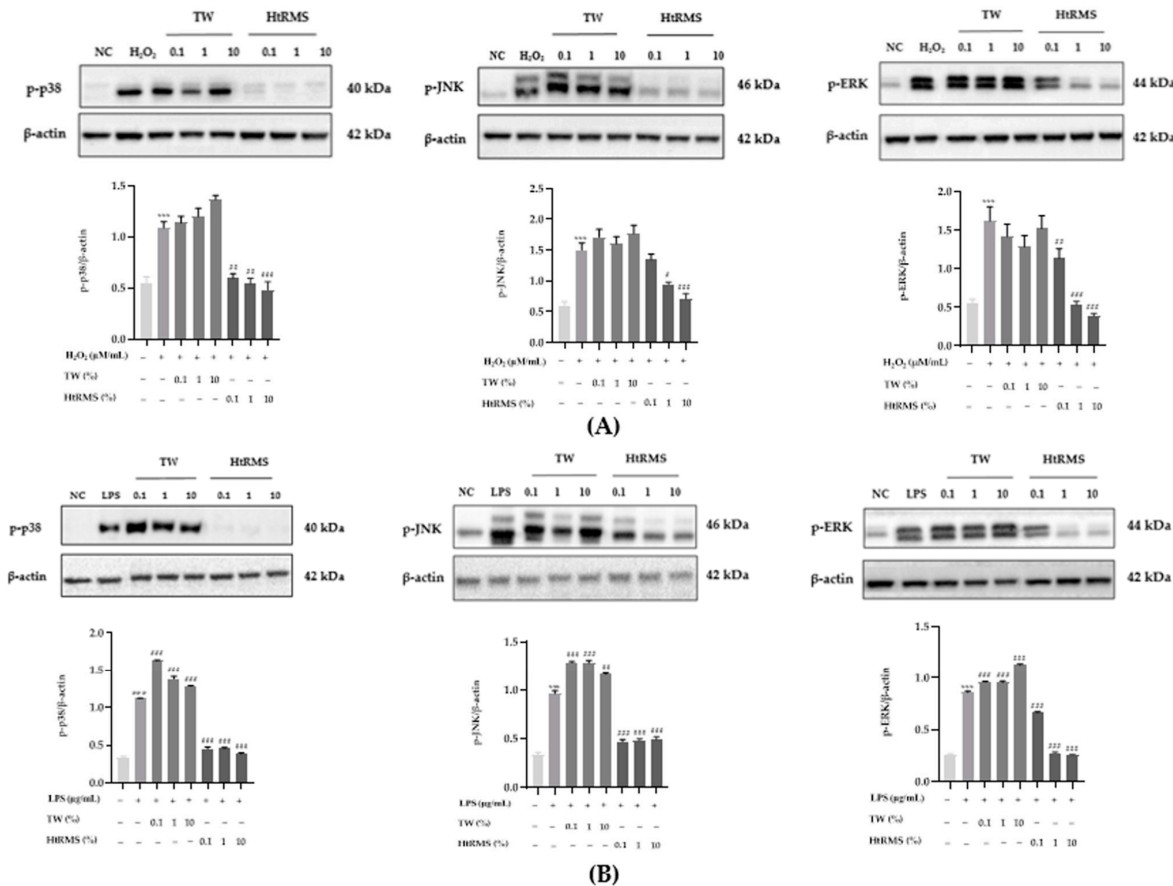

**Figure 6.** Effects of HtRMS on p-p38, p-JNK, and p-ERK of $H_2O_2$- and LPS-induced murine macrophage cells. The cells were treated with HtRMS and TW (0.1, 1, and 10%) after the induction of OS by $H_2O_2$ (**A**) and LPS (**B**), respectively. Data values are stated as mean $\pm$ standard error of the mean of fold change, compared to the control group ($n = 3$). The statistical significance was analyzed via two-way ANOVA. ($-$): non-treatment, (+): treatment. ***: $p < 0.001$ vs. normal control. #: $p < 0.05$, ##: $p < 0.01$, and ###: $p < 0.001$ vs. $H_2O_2$ or LPS only. HtRMS: high-temperature-roasted salt; TW: tap water; $H_2O_2$: hydrogen peroxide; LPS: lipopolysaccharide.

## 4. Discussion

This study evaluated the antioxidative effect of HtRMS in a murine macrophage cell line, RAW 264.7. Compared to the properties of TW [11], HtRMS is enriched in mineral contents (25 essential minerals) and has higher $H_2$, lower ORP, and higher pH. Previous studies have also revealed that the way the minerals are administered significantly impacts their antioxidative effect [8,23,24]. Specifically, minerals such as $Zn^{2+}$, $Cu^{2+}$, and $Mg^{2+}$ are known to play an essential role in the reduction of free radicals and in pathological processes, and therefore they may possibly sustain immune defense [8,24]. Our study identified a positive active role of HtRMS in inhibiting the MAPK signaling pathway. We particularly evaluated the concentration-dependent improvement of parameters such as cell viability, antioxidant enzyme levels, and concentration of OS-related markers, such as ROS, NO, and $Ca^{2+}$, and MAPK signaling pathway-related proteins, such as p-p38, p-JNK, and p-ERK after HtRMS treatment in RAW 264.7 murine macrophage cells.

First, we evaluated the effect of HtRMS on cell viability after $H_2O_2$ or LPS induction. $H_2O_2$ is commonly used to stimulate cellular OS [6], while LPS is recognized by specific host cell receptors and activates the innate immune system through an inflammatory response [25]. As expected, cell viability decreased upon induction by $H_2O_2$ and LPS; however, the application of HtRMS in murine macrophage cells significantly rescued both $H_2O_2$- and LPS-induced cell viability (Figure 1).

To assess the antioxidative effect, we measured ROS and NO production in $H_2O_2$- and LPS-induced murine macrophage cells after HtRMS treatment. ROS levels play an important role in signal transduction and cellular physiological functions in macrophage cell lines [26–28]: ROS and NO are produced and released in response to phagocytosis involved in bacterial killing, which is implicated in inflammation and tissue damage [29,30]. However, the overproduction of ROS influences NO production and consequently leads to stress and inflammatory responses in the immune system [31,32]. Consistent with previous studies on $H_2$ and electrolysis-reduced water [33–35], our results revealed that HtRMS induced a decrease in ROS and NO levels in a concentration-dependent way (Figure 2) after $H_2O_2$ and LPS induction in murine macrophage cells. The present results suggest that HtRMS may be employed in therapeutic approaches to reduce the overproduction of harmful free radicals in murine macrophage cells. Furthermore, consistent with our ROS and NO analyses, HtRMS exhibited efficacy in enhancing antioxidant effects against $H_2O_2$-induced OS and LPS-mediated inflammation. CAT and GPx protect cells against free radicals [36]. By removing them, thereby preventing and reducing oxidation-induced cell damage [37,38]. The protective effects of antioxidants are still being studied globally [38]. Evidently, cells treated with HtRMS mediated CAT and GPx levels after $H_2O_2$ and LPS induction exhibiting its antioxidant capacity (Figure 3A). It is established that antioxidants contribute to the reduction of oxidative effectors, such as ROS and NO, which are associated with inflammation in macrophage cells [2,33]. These results were consistent with the findings of previous studies, wherein electrolyzed reduced water and $H_2$ also increased antioxidant levels in in vivo models [39–42]. However, further studies are necessary to identify the mechanism through which HtRMS acts as an antioxidant to comprehensively understand and apply its clinical significance.

Moreover, we assessed intracellular $Ca^{2+}$ levels. $Ca^{2+}$ is a ubiquitous intracellular messenger that controls diverse cellular functions [43], wherein $Ca^{2+}$ becomes a mediator of cell distress, but can even be toxic, if its concentration and movement inside the cell are not regulated carefully [44]. Among these, one study revealed that intracellular $Ca^{2+}$ concentration activates the MAPK signaling pathway [45]. The present results showed that HtRMS salt protected the cells from high $Ca^{2+}$ concentrations. Therefore, our results suggest that HtRMS treatment for 24 h can diminish ROS and NO production and reduce $Ca^{2+}$ levels. Consistent with our results, several studies have proven that minerals releasing $H_2$ at high concentrations have beneficial biological effects on $Ca^{2+}$ responses [2].

Finally, to better understand the therapeutic effect of HtRMS against $H_2O_2$- and LPS-induced OS responses, the present study focused on the MAPK signaling pathway, which is

one of the three major pathways involved in early OS response and cell survival, including p-p38, p-JNK, and p-ERK [46–48]. In addition, our Western blot results indicated that HtRMS inhibited $H_2O_2$- and LPS-induced MAPK signaling pathways, including p-p38, p-JNK, and p-ERK. This suggests that HtRMS significantly reduced the OS response via regulation of the MAPK signaling pathway.

## 5. Conclusions

Overall, our results suggest that HtRMS can act as an antioxidant and may further protect the cell from the OS process through the regulation of the MAPK signaling pathway. This might be the first evidence of the antioxidative effect of HtRMS in murine macrophage cells. This study suggests the potential application of HtRMS to combat stress-related disorders. Moreover, our results suggest that fortified alkaline water can be developed using HtRMS, which evidently showed its antioxidative effect. However, there are some limitations in the current study. First, the study only presented results with the antioxidative effects of HtRMS. However, it is unclear which factor of salt, either combination of minerals or alkalinity, or both has antioxidative effects. Second, this study is limited to the use of only one macrophage cell line (RAW264.7). Therefore, further research is needed with different types of organ-specific cell lines in addition to in vivo and clinical studies. Third, we only explored the MAPK signaling pathway in this study, to confirm the antioxidative effects. Hence, other signaling pathway studies are necessary to fully claim the molecular mechanism of the antioxidative effect of this salt product.

**Author Contributions:** Conceptualization, H.-T.J.; writing—original draft preparation, T.T.T.; writing—review and editing, K.-J.L., A.F., S.-H.Y. and J.B.; methodology T.T.T., A.F., S.S. and M.H.R.; data curation, T.T.T., A.F., C.-S.K.; supervision, H.-T.J. All authors have read and agreed to the published version of the manuscript.

**Funding:** The research received no external funding.

**Institutional Review Board Statement:** Not applicable.

**Informed Consent Statement:** Not applicable.

**Data Availability Statement:** The data presented in this study are available in the article (tables and figures).

**Acknowledgments:** This research was supported by Biocera Co., Ltd., Republic of Korea.

**Conflicts of Interest:** The authors declare no conflict of interest.

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
