# Peer review of "Development of Alkaline Reduced Water Using High-Temperature-Roasted Mineral Salt and Its Antioxidative Effect in RAW 264.7 Murine Macrophage Cell Line"

_processes, doi:10.3390/pr9111928_

Round 1

Reviewer 1 Report

  1. I would like to know if there are any study limitations?  If it is possible, I would like the authors write them at the end of the article.

Author Response

Dear Reviewer,

The authors would like to take this opportunity to express our sincere gratitude  for the insightful comment. We would also like to thank you for allowing us to resubmit a revised copy of the manuscript. Please see the attachment for the point-by-point response. Thank you very much.

Sincerely,
Thuy Thi Trinh 

Reviewer 2 Report

The manuscript entitled “Development of Alkaline Reduced Water Using High-temperature Roasted Mineral Salt and its Anti-oxidative Effect in RAW 264.7 Murine Macrophage Cell Line” evaluates the anti-oxidative effect of HtRMS in a murine macrophage cell line, RAW 264.7. The results of this study suggest that HtRMS can act as an antioxidant and may protect the cell from the OS process through the regulation of the MAPK signalling pathway.

I have no hesitation in accepting this manuscript for publication once the below minor points have been attended to.

  1. Line 47: correct “lead” in “leads”.
  2. Line 176: delete remove space after 525 nm emission.
  3. Lines 223, 226, 227: write the numbers in subscript in H2O2.
  4. Lines 39, 49, 85, 94, 113, 205, 207, 210, 317, 320, 356, 362, 365, 367, 370: correct “signaling“ in “signalling”.

Author Response

Dear Reviewer,

The authors would like to take this opportunity to express our sincere gratitude  for the careful checking of the manuscript. We would also like to thank you for allowing us to resubmit a revised copy of the manuscript. Please see the attachment for the point-by-point response. Thank you very much.

Sincerely,
Thuy Thi Trinh 
